# A Precision Medicine Approach to Optimize Modulator Therapy for Rare CFTR Folding Mutants

**DOI:** 10.3390/jpm11070643

**Published:** 2021-07-07

**Authors:** Guido Veit, Tony Velkov, Haijin Xu, Nathalie Vadeboncoeur, Lara Bilodeau, Elias Matouk, Gergely L. Lukacs

**Affiliations:** 1Department of Physiology, McGill University, Montréal, QC H3G 1Y6, Canada; haijin.xu@mcgill.ca; 2Department of Pharmacology & Therapeutics, School of Biomedical Sciences, Faculty of Medicine, Dentistry and Health Sciences, The University of Melbourne, Melbourne 3010, Australia; tony.velkov@unimelb.edu.au; 3Research Center, Institut Universitaire de Cardiologie et de Pneumologie de Québec, Université Laval, Québec, QC G1V 4G5, Canada; nathalie.vadeboncoeur@criucpq.ulaval.ca (N.V.); lara.bilodeau@criucpq.ulaval.ca (L.B.); 4Adult Cystic Fibrosis Clinic, Montreal Chest Institute, McGill University, Montréal, QC H4A 3J1, Canada; elias.matouk@mcgill.ca; 5Department of Biochemistry, McGill University, Montréal, QC H3G 1Y6, Canada

**Keywords:** cystic fibrosis, cystic fibrosis transmembrane conducatance regulator (CFTR), CFTR modulator combination, primary human nasal epithelia, CFTR missense mutations, precision medicine

## Abstract

Trikafta, a triple-combination drug, consisting of folding correctors VX-661 (tezacaftor), VX-445 (elexacaftor) and the gating potentiator VX-770 (ivacaftor) provided unprecedented clinical benefits for patients with the most common cystic fibrosis (CF) mutation, F508del. Trikafta indications were recently expanded to additional 177 mutations in the CF transmembrane conductance regulator (CFTR). To minimize life-long pharmacological and financial burden of drug administration, if possible, we determined the necessary and sufficient modulator combination that can achieve maximal benefit in preclinical setting for selected mutants. To this end, the biochemical and functional rescue of single corrector-responsive rare mutants were investigated in a bronchial epithelial cell line and patient-derived human primary nasal epithelia (HNE), respectively. The plasma membrane density of P67L-, L206W- or S549R-CFTR corrected by VX-661 or other type I correctors was moderately increased by VX-445. Short-circuit current measurements of HNE, however, uncovered that correction comparable to Trikafta was achieved for S549R-CFTR by VX-661 + VX-770 and for P67L- and L206W-CFTR by the VX-661 + VX-445 combination. Thus, introduction of a third modulator may not provide additional benefit for patients with a subset of rare CFTR missense mutations. These results also underscore that HNE, as a precision medicine model, enable the optimization of mutation-specific modulator combinations to maximize their efficacy and minimize life-long drug exposure of CF patients.

## 1. Introduction

Cystic fibrosis (CF), one of the most prevalent life-threatening monogenetic diseases, is caused by loss-of-function mutations in the CF transmembrane conductance regulator (CFTR) gene [1]. The CFTR proteins forms an anion channel expressed at the apical surface of various epithelia, which is involved in the ion composition, pH and volume regulation of the luminal fluid in the airway, intestine and biliary, as well as pancreatic duct [1]. CFTR is a member of the ABC transporter superfamily and consist of two homologous halves, each containing a membrane spanning domain (MSD) and a nucleotide binding domain (NBD) that are connected by a unique regulatory domain [2].

More than 2100 different mutations are known in the CFTR gene, of which only a minority has been studied in detail (cystic fibrosis mutation database: http://www.genet.sickkids.on.ca/app, accessed on 6 July 2021; CFTR2 project: http://cftr2.org, accessed on 6 July 2021) [3]. CFTR mutants have been classified according to their cellular phenotype into expression (class I), folding (class II), gating (class III), conductance (class IV), quantity (class V) and peripheral stability defect (class VI) associated mutations [4,5]. However, many mutations lead to multiple defects, justifying the introduction of combinatorial categories to guide the development of mutant-specific combination therapy with CFTR modulators [6].

So far four CFTR modulators have been FDA approved, which belong to two categories—gating potentiators and folding correctors [7,8]. The only approved potentiator ivacaftor (VX-770) was first developed to partially restore function of CFTR with the archetypal gating mutation G551D, which impairs ATP-binding to the second ATP binding pocket at the NBD1-NBD2 interface [9,10]. VX-770 leads to a substantial clinical benefit in patients with at least one allele carrying the G551D mutations and, since its introduction in 2012, several additional mutants have been approved for ivacaftor therapy [11,12].

The most common mutation F508del, present in ~80% of CF patients on at least one allele, impairs processing, plasma membrane (PM) expression and stability, as well as gating of the channel [13,14,15]. The mutation leads to a folding defect of the NBD1, NBD1-MSDs, as well as misfolding of the NBD2 [16,17,18]. We and others showed that pharmacological chaperones targeting distinct structural defects of the NBD1-MSDs (type I), NBD2 (type II) and NBD1 (type III) are additive in their correction efficacy of the F508del-CFTR [19,20,21,22,23]. The first approved correctors, lumacaftor (VX-809) and tezacaftor (VX-661), follow a type I mechanism and in combination with VX-770 lead to a moderate clinical benefit in patients carrying one or two alleles with the F508del mutation [20,24,25,26]. 

Recently the modulator combination Trikafta (elexacaftor-tezacaftor-ivacaftor) was approved that contains the newly developed corrector elexacaftor (VX-445), which follows a type III mechanism [27,28]. VX-445 in combination with VX-661 provides at least additive rescue of the F508del-CFTR folding defect, leading to an unprecedented clinical benefit for patients with one or two F508del containing alleles [27,29,30]. This can be partly attributed to the VX-445 co-potentiator activity that is additive to that of the VX-770 for F508del-CFTR, as well as to other missense mutations associated with a gating defect [31,32]. In agreement with the proposed allosteric folding and misfolding model of wild-type and mutant CFTR, respectively, numerous missense mutations located throughout the channel are, at least partially corrected by the VX-661 + VX-445 corrector combination [19,27]. Jointly, these results support the Trikafta approval for additional 177 mutations (Vertex press release: https://investors.vrtx.com/news-releases/news-release-details/vertex-announces-fda-approvals-trikaftar, accessed on 6 July 2021).

Here we addressed whether Trikafta administration over a minimalistic drug combination with reduced life-long burden of drug exposure would ensure significant benefit as the standard of care for rare CF folding mutations. To this end we investigated the biochemical and functional correction efficacy of VX-661 + VX-445 in relation to that of type I correctors alone in selected mutants in both, the bronchial epithelial cell line (CFBE14o-) and patient-derived human nasal epithelia (HNE). The results suggest that the approach may serve as a precision medicine tool to minimize the necessary and sufficient modulator combinations in a mutation-specific manner.

## 2. Materials and Methods

### 2.1. CFTR Modulators

CFTR modulators VX-770, VX-661, VX-809, FDL169 were purchased from Selleckchem. VX-445 and ABBV-2222 were synthesized as described before [27]. 4172 and 3151 were acquired from Life Chemicals and their function as CFTR correctors has been described [19].

### 2.2. Cell Lines

We previously described the generation of CFBE41o- (a gift from D. Gruenert, University of California, San Francisco) cells lines expressing inducible P67L-, L206W- [19], S549R- [33] and WT-CFTR [34] all with a 3HA-tag in the fourth extracellular loop [35]. These cell lines were grown in MEM medium containing 10% fetal bovine serum, 10 mM 4-(2-hydroxyethyl)-1-piperazineethanesulfonic acid (HEPES) and 2 mM L-glutamine. To induce the expression of CFTR variants the cells were treated for ≥3 days with 500 ng/mL doxycycline.

### 2.3. Human Nasal Epithelia

HNE cell isolation from tissue collected by scrape biopsy was performed as described [36]. HNE were expanded in presence of irradiated feeder cells in Rock inhibitor containing medium F, a process termed conditional reprogramming [37]. Functional measurements were performed in filter grown differentiated HNE. For this purpose, HNE were seeded at a density of 5 × 10^5^ cell/filter on 1.12 cm^2^ Snapwell filter supports (Corning) and differentiated under air-liquid interface by culturing in PneumaCult-ALI medium (Stemcell Technologies) for ≥three weeks.

### 2.4. PM Density Measurement

To determine the PM density of 3HA-tagged CFTRs in CFBE, a cell surface enzyme-linked immunosorbent assay (ELISA) using mouse monoclonal anti-hemagglutinin (HA) antibody (Biolegend, clone 16B12, order no. 901524, 1:2000) was applied [14]. Cells were treated with compounds or 0.2% DMSO (vehicle control) for 24 h at 37 °C in full medium. PM density values were normalized with cell viability determined by alamarBlue Assay (Invitrogen) and related to WT-CFTR by normalizing with CFTR mRNA abundance measured by qPCR as previously [19].

### 2.5. Peripheral Stability Measurement and Immunoblotting

To measure the stability of the complex-glycosylated form of CFTR variants, CFBE cells were treated with 100 μg/mL cycloheximide (CHX) for 0–7.5 h at 37 °C in full medium. Cells were lysed in RIPA buffer supplemented with protease inhibitors (Complete, Roche) followed by removal of insoluble material, protein concentration determination, immunoblotting and CFTR detection using the mouse monoclonal anti-hemagglutinin (HA) antibody as described before [14].

### 2.6. Short-Circuit Current Measurement

Short-circuit current (I_sc_) measurement of polarized HNE was performed as described [19,27]. HNE were treated with CFTR modulators or 0.2% DMSO (vehicle control) for 24 h at 37 °C in serum-free PneumaCult-ALI medium (StemCell Technologies). Snapwell filters were mounted in Ussing chambers (Physiologic Instruments) in Krebs-bicarbonate Ringer (KBR) buffer (140 mM Na^+^, 120 mM Cl^−^, 5.2 mM K^+^, 25 mM HCO_3_^−^, 2.4 mM HPO_4_, 0.4 mM H_2_PO_4_, 1.2 mM Ca^2+^, 1.2 mM Mg^2+^, 5 mM glucose, pH 7.4), which was mixed by bubbling with carbogen (95% O_2_ and 5% CO_2_). To generate a basolateral-to-apical chloride gradient NaCl was replaced with 115 mM Na^+^ gluconate in the apical buffer and the I_sc_ was determined in the presence of 100 μM amiloride. The transepithelial voltage was clamped at 0 mV (VCC MC8 multichannel voltage/current clamp, Physiologic Instruments) after compensating for voltage offsets and current and resistance were recorded at 37 °C with the Acquire and Analyze package (Physiologic Instruments).

### 2.7. Statistics

Results are presented as mean ± SEM with the number of experiments indicated. Statistical analysis was performed by two-tailed Student’s *t*-test with the means of at least three independent experiments and the 95% confidence interval was considered significant.

## 3. Results

### 3.1. Identification of Mutants with High Responsiveness to Type I Correctors

To determine the relative correction efficacy of the type I corrector VX-661 and the type III corrector VX-445, we measured the PM density of 13 CF-causing mutations that are associated with folding defects in the CF bronchial epithelial cell line CFBE41o- (CFBE). These mutations, distributed throughout CFTR, lead to various reductions of the mutant PM densities relative to the wild-type (WT) after normalization for mRNA expression (Appendix A). The CFTR PM density changes ranged from very severe loss (<5% of the WT) for S13F, E92K, S492F, F508del, V520F, L1077P, M1101K, and N1303K to relatively mild reduction (>20% of the WT) for R31C, R352G and S549R, as reported previously [19,27]. Apart from S492F, V520F and M1101K, which were resistant to correction with VX-661, the processing defect of all mutations was partially reversed by both correctors to various extend (Appendix A). To identify mutations that exhibit preferential responsiveness to type I correctors, the ratio of the relative correction efficacy of VX-661 and VX-445 was calculated (Figure 1A). Mutations in the MSD1 and the proximal N-terminus were more responsive to VX-661 than to VX-445 (Figure 1A), consistent with their high responsiveness to, and the stabilization of the MSD1 and CFTR N-terminal half by VX-809 (Figure 1B) [38,39,40,41]. Mutations in the other CFTR domains showed partial selectivity for correction with VX-445, with the exception of the S549R in the NBD1, which could be nearly completely corrected by either corrector (Figure 1A and Appendix A). Since VX-661 treatment resulted in >50% of WT PM expression, the highest absolute correction of P67L, L206W and S549R variants, we chose these mutants to test whether the addition of VX-445 would provide additional benefit.

### 3.2. Cooperative Correction of the Mature Protein and PM Expression by VX-661 + VX-445 Corrector Combination

Similar to monitoring the PM density, the folding defect and its correction can be assessed by quantifying the abundance of the mature, complex-glycosylated form (band C) of CFTR mutants relative to that of the WT, after normalizing for the mRNA abundance [19]. The type I correctors VX-809 and VX-661 resulted in a substantial increase in the band C abundance of P67L-, L206W- and S549R-CFTR, which for P67L and L206W could be further increased by VX-661 + VX-445 treatment (Figure 1B). Similar results were obtained by PM density measurement for P67L- and L206W-CFTR, and the small increase in S549R-CFTR PM density upon addition of VX-445 to the type I corrector treatment reached significance in this assay (Figure 1C). Since the PM density assay allows for higher-throughput measurements, we also determined the effect of the type I correctors ABBV-2222 and FDL-169, which are in clinical development [42,43]. Similar to its effect on E92K-CFTR [27], ABBV-2222 exhibited higher efficacy to correct the folding defect of P67L- and L206W-CFTR in comparison to other type I correctors (Figure 1C). Co-treatment of all tested type I correctors with VX-445 resulted in a small, but significant, increase in the PM density of P67L-, L206W- and S549R-CFTR (Figure 1C). 

To test whether the cooperative correction can be accounted for by discrete mechanism of actions on distinct structural defects, the additivity or synergy of the two corrector classes was analyzed by combinatorial profiling. To this end, the rescue efficacy of corrector pairs was compared to their theoretical additivity, based on their individual efficacy (Appendix A). The profiles of their interaction with other preclinical correctors enabled us to cluster compounds with similar mechanisms. Cluster analysis for P67L-, L206W- and S549R-CFTR indicated that the type I correctors clustered together, VX-445 clustered together with the type III corrector 4172, and the type II corrector 3151 forms its own category with variable distance to the other two clusters, similar to the published data for F508del and some missense mutations [27] (Figure 1D). These results confirm the mechanistic classification of the correctors as proposed earlier [19,20,27] and show that mutations, which are highly responsive to stabilization of the MSD1 and MSD1-NBD1, can be further corrected by correctors targeting the NBD1 or NBD2. This is in agreement with a recent study reporting that P67L impairs folding of downstream domains including the NBD1 and NBD2 [44].

### 3.3. Correction of the Peripheral Stability Defect of P67L-, L206W- and S549R-CFTR

The folding defect introduced by the P67L, L206W and S549R mutations may reduce the trafficking of the channel to the plasma membrane, stability of the mature protein or combination of both defects. Severely reduced maturation rate of L206W and the VX-809-mediated folding correction of P67L and L206W have been shown [38,39,45], therefore we focused on determining the stability of the post-Golgi, complex-glycosylated form of the mutants. As determined by cycloheximide (CHX) chase and immunoblotting, presence of the P67L or L206W mutation reduced the half-life of complex-glycosylated CFTR to ~2.5 h and ~5 h, respectively, while the S549R only introduced a minor stability defect (Figure 2A,B). VX-809 or VX-661 partially correct the P67L-CFTR and almost completely corrected the L206W-CFTR stability defect. Co-treatment with VX-445, however, did not provide significant additional benefit (Figure 2A,B).

### 3.4. Some Patients with the L206W Mutation Exhibit Progressive Lung Function Decline

To analyze the effect of CFTR modulators on the mutant function in patient-derived airway epithelia, we identified and subsequently collected human nasal epithelia (HNE) form nine CF patients carrying the P67L, L206W or S549R mutation. Since these are rare mutations with an allelic frequency of 0.17%, 0.23% and 0.065% for P67L, L206W and S549R (CFTR2 database, www.cftr2.org, accessed on 6 July 2021), respectively, all patients are compound heterozygous with F508del as the most frequent mutation on the second allele.

The L206W mutation leads to a mild CF phenotype [45], which is exemplified by a low incidence of pancreatic insufficiency, rare chronic colonization with Pseudomonas aeruginosa and only moderate elevation of sweat chloride (CFTR2 database, www.cftr2.org, accessed on 6 July 2021). In our small patient cohort, the average age at the time of HNE collection was 54.3 ± 10.3 years (range 33–64 years) and the forced expiratory volume in 1s (FEV1) % predicted 70% ± 29% (range 39–107%), indicative for the variable disease phenotype. However, a subset of these patients had substantial airflow obstruction (FEV1% predicted <50%) and exhibited a progressive loss of lung function over time (Figure 3A). Consistent with the association of the P67L mutation to a milder phenotype [46], the lung function of the compound heterozygous F508del/P67L patient remained stable (FEV1% predicted >50%) over several years (Figure 3B). Due to the pronounced gating defect the S549R mutation is associated to a severe CF phenotype [47], however the F508del/S549R patient in this study was able to maintain a FEV1% predicted >80% after the onset of ivacaftor therapy at age 18 (Figure 3B). These results underscore the need for modulator therapy even in some patients carrying mutations that are generally associated with a milder phenotype.

### 3.5. Functional Correction of P67L-, L206W- and S549R-CFTR in Human Nasal Epithelia

The HNE harboring P67L-, L206W or S549R-CFTR were expanded by conditional reprogramming and differentiated at air-liquid interface as reported [36]. In HNE from seven patients with one allele of L206W the mean forskolin-activated short-circuit current (I_sc_) was 6.6% relative to that of WT-CFTR in HNE isolated from 10 donors [27], which could be increased to 27.5% and 25.8% by correction with VX-809 and VX-661, respectively (Figure 4A,B). Under activation conditions with saturating forskolin concentrations, the acute or chronic treatment with VX-770 did not significantly increase the I_sc_ (Figure 4A,B). For comparison, Trikafta treatment increased the F508del channel function in homozygous HNE to ~60% of the WT [27].

A large inter-donor variation in the forskolin stimulated I_sc_ in compound heterozygous L206W-CFTR HNE was observed both before and after correction with VX-809 or VX-661, similar as noted for homozygous ΔF508-CFTR HNE [19]. Concordantly, after expressing the corrected I_sc_ as percentage of the basal channel activity lower variance between HNE from individual patients was observed (Appendix A). These observations led us to investigate whether the basal channel activity is correlated to the level of correction in individual HNE. Both parametric (Pearson correlation coefficient) and non-parametric (Spearman’s rank correlation coefficient) correlation analysis indicated a highly significant correlation between the forskolin-activated basal I_sc_ and the VX-809 or VX-661 corrected I_sc_ (Figure 4C), similar to the correlation between the VX-809 corrected and basal CFTR activity observed in a panel of rare CFTR genotypes in HNE [48]. 

Next, we focused on the correction of L206W in compound heterozygous HNE carrying a splice site mutation on the second allele, which leads to no or substantially reduced protein expression. Similar to the global analysis of L206W carrying HNE, VX-809 or VX-661 resulted in a significant correction of the forskolin-activated I_sc_ in these HNE, which could not be further increased by VX-770 in either the partially or completely activated channels (Figure 4D and Appendix A). Importantly, the addition of VX-445 led to heterogenic effects and only further improved the VX-661 corrected I_sc_ in one of the two tested HNE activated with saturating forskolin concentrations (Figure 4D). In contrast, VX-445 increased the I_sc_ in both HNE in which CFTR was partially activated with 100 nM forskolin (Appendix A).

We also investigated the corrector response of P67L/F508del and S549R/F508del compound heterozygous HNE. The CFTR function in P67L/F508del HNE was significantly corrected by VX-809 and VX-661 (Figure 5A,B). The functional correction of the partially or fully activated P67L/F508del CFTR was further augmented by acute, but not by chronic addition of VX-770 (Figure 5A,B and Appendix A), likely due to destabilization of both, corrected P67L and F508del proteins, by chronic VX-770 exposure as reported before [36,49]. CFTR function in these cells was also partially corrected by VX-445 and VX-661 + VX-445 co-treatment significantly increased the correction efficacy. Single corrector treatment led only to minor correction of the CFTR function in S549R/F508del HNE, which was significantly increased by both acute and chronic VX-770 exposure, consistent with the gating defect of the S549R mutant (Figure 5A,B) [33,50]. Dual corrector treatment, however, did not further increase the functional correction efficacy (Figure 5A,B). While the contribution of the F508del allele to the functional correction in these HNE is not established, the responses to correctors were similar to the biochemical results obtained in CFBE cells (Figure 1).

A weak correlation between the FEV1% predicted of F508del homozygous CF patients and the forskolin-induced swelling of corresponding patient-derived intestinal organoids has recently been reported, suggesting that differences in low CFTR residual function may contribute to clinical heterogeneity in F508del homozygous patients [51]. In case of the compound heterozygous P67L-, L206W or S549R-CFTR patients, however, we did not observe significant correlation between the basal channel function in HNE and the FEV1% predicted, which may be explained by the lower sample size in our study or the higher average age that augments the influence of environmental factors (Appendix A).

## 4. Discussion

The approval of Trikafta, the combination of the folding correctors VX-661 + VX-445 and the gating potentiator VX-770, was recently expanded to 177 additional CFTR mutations located throughout the channel based on in vitro data in FRT cells, so far without formal publication of these results (Vertex press release: https://investors.vrtx.com/news-releases/news-release-details/vertex-announces-fda-approvals-trikaftar, accessed on 6 July 2021) [52]. Here we are investigating three of these mutants, which were selected based on their high responsiveness to type I correctors, to analyze their responsiveness to the individual and combinations of CFTR modulators with the goal to achieve the highest functional correction efficacy with the least number of compounds. This goal is desirable since CFTR modulators can have adverse drug–drug interactions as has been shown for VX-809, which induces the cytochrome P450 variant CPY3A4 that is the metabolizing enzyme for VX-770, thus reducing the VX-770 plasma concentration [53]. VX-770 also has been shown to destabilize F508del-CFTR and some rare missense mutants, which attenuates the corrector efficacy [36,49].

In CFBE cells, treatment of P67L-, L206W- and S549R-CFTR with type I correctors corrected the PM density and complex-glycosylated form expression to ~30–80% of the WT in agreement with published results [19,27,39]. The notable exception was treatment of P67L- and L206W-CFTR with the corrector ABBV-2222 [42] that resulted in ≥WT PM densities, indicating variable efficacies within the group of type I correctors. Co-treatment with the type III corrector VX-445 substantially increased the correction efficacy of P67L- and L206W-CFTR but had a less pronounced effect on S549R-CFTR. Consistent with the tight correlation between corrector effects in CFBE and HNE cells [19], similar trends were observed for the functional measurements in HNE cells. The I_sc_ of compound heterozygous HNE with one allele of P67L-, L206W- or S549R-CFTR was partially corrected by VX-809 or VX-661, which in P67L/F508del and in one of the two L206W/splice mutations containing HNE could be augmented by co-treatment with VX-445. Under conditions of CFTR activation by saturating concentrations of forskolin, VX-770 did not further increase the I_sc_ in L206W-CFTR containing HNE. In P67L containing HNE the increase in I_sc_ by acute VX-770 addition was negated by chronic VX-770 exposure likely due to destabilization of the corrected channel as reported before [36]. Since the level of channel activity and extend of CFTR phosphorylation in the lung tissue are not known, the modulator responses were also examined in HNE partially activated with 100 nM forskolin, which confirmed that VX-661 + VX-445 are sufficient for the functional correction of the P67L and L206W mutants. Consistent with clinical responses [47], the I_sc_ was significantly corrected by VX-770 in S549R-CFTR containing HNE, which could be augmented by type I correctors but not by co-treatment with VX-445. The I_sc_ measurements in L206W-containing HNE also showed that the residual channel function, which depends on individual’s genetic and epigenetic determinant factors, is correlated to the corrector response. Thus, assuming that the CFTR function in HNE is a therapeutic marker for clinical responsiveness to CFTR modulators as has been suggested for F508del-CFTR [48,54] and a variety of other mutations [55], measurement of the basal channel function could predict the extend of a patient’s clinical responsiveness.

We recently proposed the concept of allosteric corrector combinations, which is based on the posttranslational completion of the CFTR cooperative domain-folding and the coupled domain-misfolding of F508del and other folding mutations [18,20,56,57]. It predicts that the localized stabilization of CFTR domains by correctors of distinct folding defects will be propagated to distant regions of the channel and, thereby, combinations of correctors targeting different CFTR domains may synergistically stabilize a variety of mutants [19,27]. However, with mutants that are highly corrected by type I correctors the fractional benefits of additional correctors targeted to different domains is diminishing, which could explain the low responsiveness of S549R-CFTR to co-treatment with VX-445.

If the results in HNE can be extrapolated to the clinic, VX-661 + VX-770 (Symdeko) or VX-809 + VX-770 (Orkambi) will provide more efficacious restoration of the channel function in patients carrying the S549R mutation in comparison to VX-770 (Kalydeco) treatment alone and comparable to Trikafta. Patients carrying the P67L- and L206W-CFTR mutations could benefit from the VX-661 + VX-445 corrector combination, which is so far not marketed separately, but may not necessarily profit from the addition of VX-770. This study adds to the growing number of publications, in which patient-derived HNE are used to study the modulator responses of rare CFTR mutants [27,31,32,33,58,59,60,61,62]. Our results suggest that HNE can not only be used to identify modulator responsive mutants, but also to optimize the modulator combinations for rare mutants. Thus, HNE may serve as a precision medicine tool to optimize the modulator therapy regiment for CF patients with rare mutations.

## Figures and Tables

**Figure 1 jpm-11-00643-f001:**
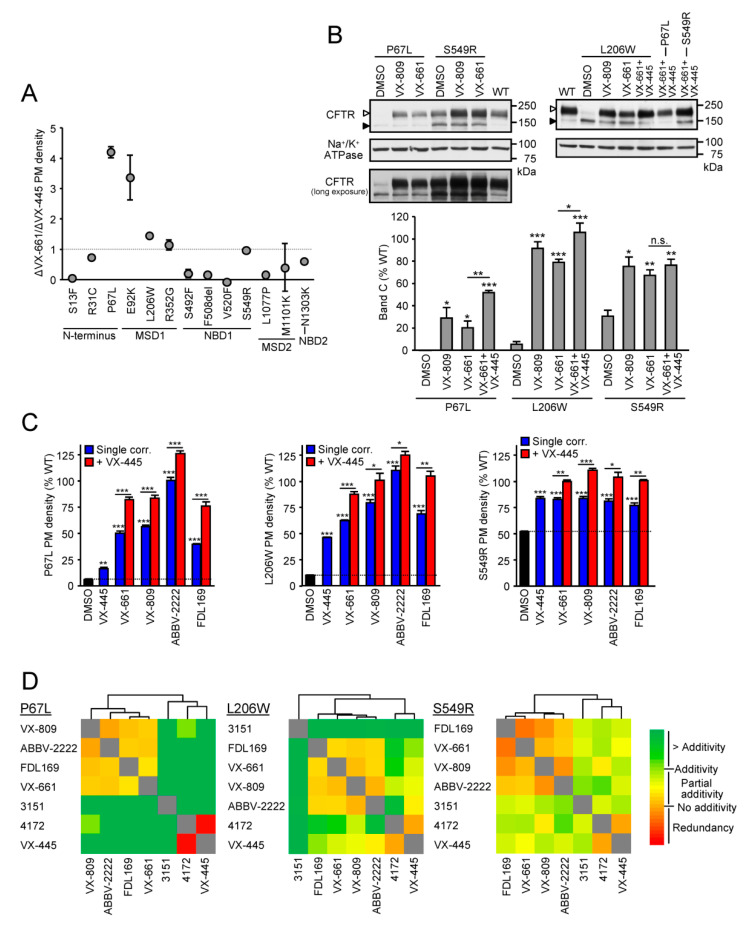
P67L-, L206W- and S549R-CFTR are highly responsive to type I correctors. (**A**) Ratio of the relative responsiveness of the indicated CFTR mutants expressed in CFBE to VX-661 (3 µM, 24 h, 37 °C) and VX-445 (2 µM, 24 h, 37 °C) determined by PM density measurement (*n* = 3). The variable absolute responses to single correctors expressed as percentage of WT-CFTR PM density are depicted in Appendix A. (**B**) Immunoblot (upper panel) of CFBE expressing inducible CFTR mutants with an extracellular 3HA tag. The cells were incubated for 24h with DMSO, VX-809 (3 µM), VX-661 (3 µM) or VX-661 + VX-445 (2 µM). CFTR was visualized with anti-HA antibody, and anti-Na^+^/K^+^-ATPase served as loading control. The empty arrowheads indicate the mature, complex-glycosylated CFTR (C-band), and the filled arrowhead show the immature, core-glycosylated protein (B-band). The the complex-glycosylated form (band C, lower panel, *n* = 3) was quantified by densitometry and, after normalization for CFTR mRNA abundance, is expressed as percentage of WT-CFTR. To visualize the low abundance of uncorrected P67L band C, a longer exposure is shown. (**C**) PM density of CFTR mutants after type I corrector (VX-661, VX-809, ABBV-2222 or FDL169–3 µM, 24 h, 37 °C), VX-445 (2 µM, 24 h, 37 °C) or type I + VX-445 corrector combination treatment expressed as percentage of WT-CFTR in CFBE (*n* = 3). (**D**) Heat map of the combinatorial profiling for P67L-, L206W- and S549R-CFTR established by calculating the dual corrector effect in relation to the theoretical additivity of the compounds. Combinatorial profiles were subsequently used to cluster compounds by average linkage analysis and the distance was determined by Spearman‘s rank correlation. The underlying data are depicted as bar plots in Appendix A. Data in A–C are means ± SEM of three independent experiments. * *p* < 0.05, ** *p* < 0.01, *** *p* < 0.001 by Student’s *t*-test.

**Figure 2 jpm-11-00643-f002:**
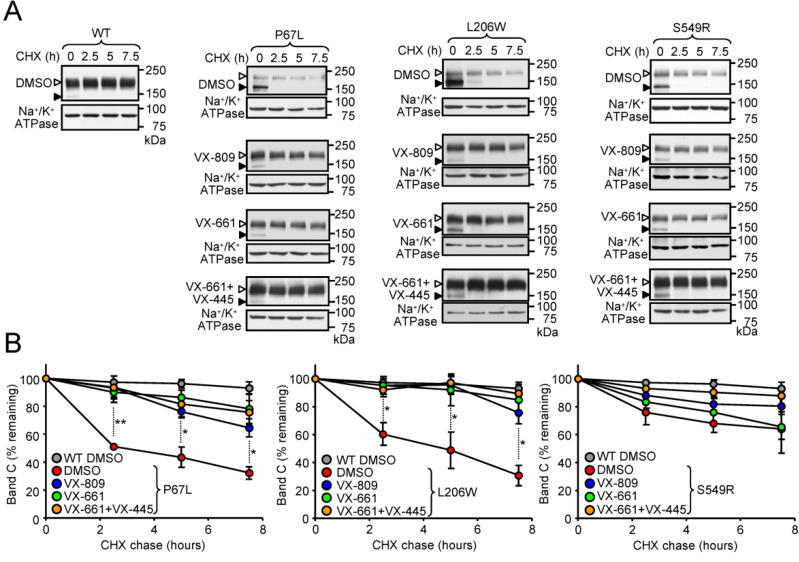
Effect of correctors on the metabolic stability of P67L-, L206W- and S549R-CFTR. (**A**) Metabolic stability of the WT and CFTR mutants was measured by CHX chase at 37 °C and immunoblotting with anti-HA antibody in CFBE cells pre-treated for 24 h with the indicated compounds. Anti-Na^+^/K^+^-ATPase served as loading control. The empty arrowheads indicate the mature, complex-glycosylated CFTR (C-band), and the filled arrowhead show the immature, core-glycosylated protein (B-band). (**B**) Remaining complex-glycosylated CFTR was quantified by densitometry and expressed as a percentage of the initial amount (*n* =3). Data in B are means ± SEM of three independent experiments. * *p* < 0.05, ** *p* < 0.01 by Student’s *t*-test.

**Figure 3 jpm-11-00643-f003:**
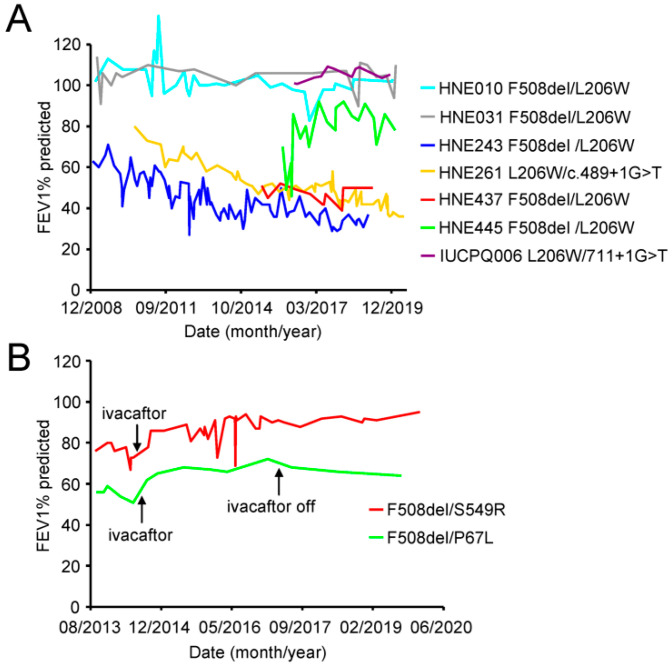
Longitudinal development of the FEV1% predicted in patients carrying one allele of the P67L-, L206W- or S549R-CFTR mutation. (**A**) Development of the FEV1% predicted over time for patients carrying the L206W mutation on one allele. (**B**) Development of the FEV1% predicted over time for patients carrying one allele of the P67L or S549R mutation.

**Figure 4 jpm-11-00643-f004:**
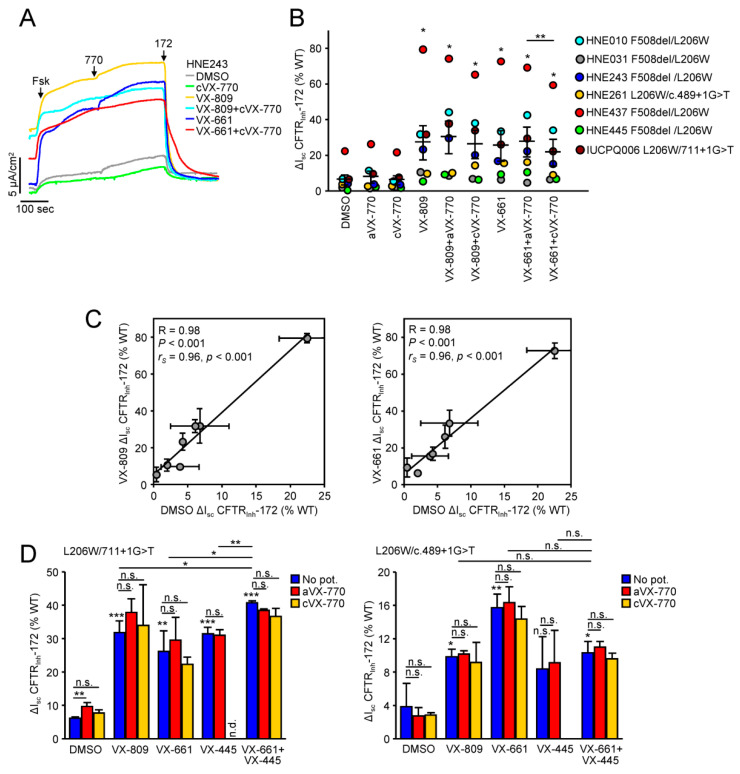
Efficacy of CFTR modulators for the functional correction of L206W-CFTR in HNE. (**A**,**B**) Effect of indicated single correctors (VX-809, VX-661-3µM, 24 h) alone or in combination with acute VX-770 (aVX-770) or chronic VX-770 (cVX-770-1 μM, 24 h) potentiation on the I_sc_ of HNE from 7 patients with one L206W-CFTR allele. Representative traces (**A**) and quantification of the CFTR_Inh_-172 inhibited current expressed as percentage of WT-CFTR currents in HNE from 10 donors (**B**). CFTR-mediated currents were induced by sequential acute addition of forskolin (Fsk, 20 μM) and VX-770 (10 μM, aVX-770) followed by CFTR inhibition with CFTR_Inh_-172 (Inh-172, 20 μM) in an intact monolayer with basolateral-to-apical chloride gradient. Black lines in B indicate mean ± SEM. (**C**) Correlation between the basal Fsk-stimulated current and the Fsk-stimulated current in VX-809 (left panel) or VX-661 corrected (right panel) HNE. R, *P*-Pearson correlation coefficient and associated *p*-value; *r_S_*, *p*-Spearman’s rank correlation coefficient and associated *p*-value. (**D**) Effect of indicated single correctors or corrector combinations (VX-809, VX-661-3µM; VX-445-2µM; 24 h) alone (no pot.) or in combination with acute VX-770 (aVX-770) or chronic VX-770 (cVX-770-1 μM, 24 h) potentiation on the I_sc_ of HNE from 2 patients with one L206W-CFTR allele and a splice site mutation on the second allele. Data in C and D are means ± SD of 3 measurements. n.s.—not significant, * *p* < 0.05, ** *p* < 0.01, *** *p* < 0.001 by Student’s *t*-test.

**Figure 5 jpm-11-00643-f005:**
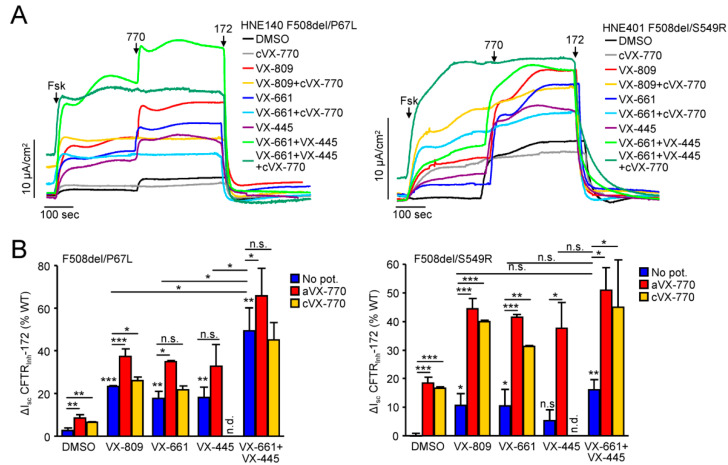
Efficacy of CFTR modulators for the functional correction of P67L- or S549R-CFTR in HNE. (**A**,**B**) Effect of indicated single correctors or corrector combinations (VX-809, VX-661-3µM; VX-445, 2µM; 24 h) alone (no pot.) or in combination with acute VX-770 (aVX-770) or chronic VX-770 (cVX-770-1 μM, 24 h) potentiation on the I_sc_ of HNE with a F508del/P67L (left panels) or F508del/S549R (right panels) genotype. Representative traces (**A**) and quantification of the CFTR_Inh_-172 inhibited current expressed as percentage of WT-CFTR currents in HNE from 10 donors (**B**). Data in B are means ± SD of 3 measurements. n.s.—not significant, * *p* < 0.05, ** *p* < 0.01, *** *p* < 0.001 by Student’s *t*-test.

## Data Availability

The data presented in this study are available in the article and Appendix A.

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
