# Peer review of "A Precision Medicine Approach to Optimize Modulator Therapy for Rare CFTR Folding Mutants"

_jpm, 2021, doi:10.3390/jpm11070643_

Round 1

Reviewer 1 Report

This manuscript by Veit et al examines the effects of various CFTR correctors and potentiators, including those that are components of the drug, Trikafta, on rescue of rare CFTR mutations in various biochemical and functional assays using a cell line and primary nasal culture models.

Major Comments:

The data are solid and convincing. A statement addressing potential differences of findings in cell lines vs primary cells should be added. How good do CFBE cells represent HNE?

Does the tag in CFTR utilized for cell surface ELISA affect CFTR stability?

Please discuss how PK considerations may affect rescue in patients. The lack of correlation of HNE in vitro data with FEV1 improvement may also be discussed.

Where were data shown in Figure 3 derived from?

Minor comments:

Figure 1B shows no immature P67L in DMSO control, however Fig. 2A DMSO P67L panel does show both strong B and weak C band.

Figure 1 B the labeling on the right side of the blot is confusing.

Fig 4A: why is there no forskolin plateau?

Page 6 line 266: what is meant by "heterogenic effect."

Please give an explanation and/or reference for all utilized modulators in the methods section (e.g, 3151 and 4172).

Supplemental figures:

S1B-D: Clarify that "Theoretical" and "calculated additivity" are she same.

S2: Why was acute VX-770 sued at 10uM that appears high. Last sentence of figure legends: is this 9 patients per mutation or altogether, please clarify.

Reviewer 2 Report

The manuscript by Veit et al. investigated the effect of the three FDA approved therapies for Cystic Fibrosis in bronchial epithelial cells and primary nasal epithelial cells bearing rare CFTR mutations (P67L, L206W and S549S). The authors performed different methods related to electrophysiology and biochemistry in the  manuscript. The paper is clearly written and has a good flow. The manuscript perfectly fits into the scope of the journal. Hence, this manuscript could be interesting for the readers and I recommend it for publication following the consideration of the minor revisions stated as follows:

Line 70: please include the following citations: 10.1124/mol.118.111799; 10.1124/mol.117.108373

Figure 1B: the authors included the WB analysis (Band C) for VX-661+VX-445 treatment but is missing the representative blot. Please include it.

Figure 2A: the authors tested the single effect of VX-661 and VX-809 and then VX-661+VX-445. Did the authors test the effect of VX-445 alone? 

Figure 4: The ussing chamber data presented, are on HNE from CF patients bearing F508del and one of the studied CFTR rare mutations. To understand how much CFTR function is rescued from the rare mutation, it will be helpful to include the F508del/F508del rescued CFTR function. The author may include the published data (from Veit et al. Nat Med) as threshold (dashed line). 

Since the HNE were from heterozygous patients, the authors should less empathize that FDA approved drugs corrects these rare CFTR variants. The authors should also acknowledge that these data need to be tested in HNE obtained from rare homozygous CF patients, even these mutations are ultra rare.
